# *Lidosomes*: Innovative Vesicular Systems Prepared from Lidocaine *Surfadrug*

**DOI:** 10.3390/pharmaceutics14102190

**Published:** 2022-10-14

**Authors:** Martina Romeo, Elisabetta Mazzotta, Ida Daniela Perrotta, Rita Muzzalupo

**Affiliations:** 1Department of Pharmacy, Health and Nutritional Sciences, University of Calabria, 87036 Arcavacata di Rende, Italy; 2Centre for Microscopy and Microanalysis (CM2), Department of Biology Ecology and Earth Sciences, University of Calabria, 87036 Arcavacata di Rende, Italy

**Keywords:** Lidocaine, surfadrug, vesicles, skin permeation, drug release

## Abstract

Lidocaine is a local anaesthetic drug with an amphiphilic structure able to self-associate, under certain conditions, in molecular aggregates playing the role of both carrier and drug. The aim of this study was to determine the optimal conditions for obtaining vesicular carriers, called lidosomes. The new formulations were obtained using both lidocaine and lidocaine hydrochloride and different hydration medias (distilled water, acid, and basic aqueous solution). Lidosomes formulations were characterized in terms of size, ζ-potential, drug retained, stability formulation, and ex vivo permeation profile. Moreover, lidosomes were incorporated in two different gel structures: one based on carboxymethylcellulose and one based on pluronic F-127 to achieve suitable properties for a topical application. Results obtained showed that lidocaine showed a better performance to aggregate in vesicular carriers in respect to hydrochloride form. Consequently, only formulations comprised of lidocaine were studied in terms of skin permeation performance and as carriers of another model drug, capsaicin, for a potential combined therapy. Lidocaine, when in form of vesicular aggregates, acted as percutaneous permeation enhancer showing better permeation profiles with respect to drug solutions. Moreover, lidosomes created a significant drug depot into the skin from which the drug was available for a prolonged time, a suitable feature for a successful local therapy.

## 1. Introduction

Painkillers are the most widely used local anaesthetics, which reversibly inhibit pain impulse transmission by blocking voltage-dependent channels.

Among them, Lidocaine (LID), or Xylocaine, is an amino-type anaesthetic largely employed in clinical practice, since its binding ability to voltage-gated sodium channels of excitable membranes results in the inhibition of action potential depolarization and the conduction of noxious stimuli to the central nervous system [1]. LID is an efficacious local anaesthetic at fast onset and low systemic toxicity. Further, LID exhibits a short action duration and antiarrhythmic activity and, thus, its administration needs caution due to these cardiovascular effects. 

LID performance could be improved by nanotechnology approaches that could increase resistance time on the skin, in order to allow the penetration into corneum stratum and pain receptors underlying desensitization [2,3,4,5]. Nanoscale drug delivery systems (DDS) have been designed as inert systems that can transport drugs at the target site with high selectivity and controllable kinetic release profile [6,7]. Various DDS, such as niosomes, liposomes, transferosomes, and solid lipid nanoparticles, have already been employed to improve LID efficacy due to the role of these systems as skin permeation enhancers able to increase drug permeability, and consequently, ensure a higher drug accumulation at the target site and thus, an improved analgesic effect [8,9,10,11]. Their better permeation ability allows for lower drug doses and could reduce cardiovascular side effects [12,13].

LID possesses an interesting amphiphilic structure that allows it to act as a surface-active drug.

The ability to aggregate in micelles depends on the drug chemical structure and the balance between drug–water interaction and hydrophobic interaction (i.e., solute–solute interaction). Shaik and co-workers [14] investigated the aggregation properties of Lidocaine Hydrochloride (LIDHCL) in water and found that at specific concentrations it formed micelles due to the establishment of hydrophobic interactions between the amide bond (CH_2_CONH) and the methyl side group of the benzene ring. 

In our previous study, we demonstrated the ability of other surfadrugs, such as Cromolyn and Diclofenac Sodium, to form at specific conditions in vesicular structures, while simultaneously playing the role of both bilayer constituent and drug [15,16]. These innovative nanosized vesicles offered several advantages, such as the ability for better skin permeation and the possibility to avoid the use of additional excipients, improving formulation biosafety, and increasing drug loading [17]. Considering these findings, we decided to study, for the first time, the aggregation properties of lidocaine as vesicle structures (lidosomes, LD) and to find the best experimental conditions to achieve nano-size carriers with physicochemical properties suitable for topical applications. LD were also included in gel created of carboxymethylcellulose (CMC) or pluronic F-127, while ex vivo permeation studies were carried out to investigate drug permeation profile. Moreover, surfadrug-based vesicles may also be used as carriers of other drugs for a potential synergic therapy. In this light, a further step forward had been taken in this work with respect to the previous study, since vesicles comprised of surfadrugs and loaded with a model drug, capsaicin (CA), were designed and their skin permeation performance was simultaneously investigated to predict their role for a potential synergic therapy. 

## 2. Materials and Methods

### 2.1. Chemicals

Lidocaine Hydrochloride monohydrate (LIDHCl), Lidocaine (LID), Capsaicin (CA), and pluronic F-127 were purchased from Sigma-Aldrich (Milan, Italy, purity 99%); Carboxymethyl cellulose was purchased from Polichimica S.R.L (Bologna, Italy); the organic solvents were supplied from Sigma-Aldrich (Milan, Italy).

### 2.2. Lidosomes Preparation

Niosomes based on LID and LIDHCl were prepared by a thin layer evaporation [18]. Accurately weighed quantities of the drug were dissolved in ethanol into a round-bottom flask. The solvent was evaporated through the application of reduced pressure and constant rotation to form a thin lipid film. The lipid film was then hydrated with 10 mL of distilled water, acid, or basic solution (pH 5.5 or 7.9, respectively) at 60 °C for 30 min, to form large multilamellar vesicles (MLV). After preparation, dispersions were left to equilibrate at 25 °C overnight. The reduction of vesicular size was achieved by sonication conducted in an ultrasonic bath at 60 °C for 30 min. The amplified mechanical vibrations generated small unilamellar vesicles (SUV) from MLVs. Subsequently, the vesicles were purified by exhaustive dialysis for 4 h in distilled water using Visking tubing (Spectra/Por^®^, cut-off 12–14 kDa), manipulated before use according to Fenton’s method [19]. After purification, formulations were stored at 4°C until the subsequent experiments.

LD loaded with CA (LD-CA) in a 9:1 molar ratio were subsequently prepared by using the same method. CA together with LID were dissolved in ethanol. The LD-CA was purified by exhaustive dialysis for 4 h using an 80:20 hydroalcoholic solution as a medium. Details on vesicles compositions are reported in Table 1. 

Furthermore, LD gel formulations were prepared to ensure they were suitable for skin application. Specifically, we decided to formulate two different gels, using two different polymers: F127 and CMC. In fact, F127 was widely known as a skin permeation enhancer, thus we compared its performance with that of an inert polymer. To this purpose, 5 mL of LD solution was added to 0.150 g of CMC or 0.900 g of block copolymer and homogenized in accordance with the procedure reported in [20].

### 2.3. Niosomal Characterization

LD diameter, size distribution, and ζ-potential were determined by Zetasizer ZS Malvern Instruments Ltd. (Malvern, U.K.) at 25 ± 0.1 °C.

After the samples were purified by dialysis, all analyses were conducted in triplicate and expressed as mean ± standard deviation.

The morphology of lidosomes was determined by Transmission Electron Microscopy (TEM) and the images were obtained with TEM ZEISS EM 10. Two drops of formulations placed on a copper grid with a nitrocellulose covering were stained with phosphotungstic acid (2%, *w*/*v*) and left to dry at room temperature before the microscopy observation.

The stability of the niosomes was evaluated by storing the formulation at 4 °C for three months and monitoring diameter, PI and **ζ**-potential initially after 15 days, and then every month. Each analysis was carried out in triplicate.

#### Determination of the Amount of LID in Vesicular Systems

The amount of LID retained in the vesicles was determined after sample purification by exhaustive dialysis. A total of 1 mL of purified and 1 mL of non-purified LD were diluted in 100 mL of ethanol, and the LID concentration was measured spectrophotometrically (Thermo Fisher scientific evolution 201/220, Waltham, MA, USA) at 262 nm corresponding to the LID wavelength. The drug content retained in the formation of vesicles structure (DL%) was calculated according to the following Equation (1):(1)$$DL\%=\left(\frac{C_{dr}}{C_{d_{i}}}\right)\times 100$$
where *C_dr_* is the initial drug concentration measured after purification process and *C_di_* is drug concentration used for the preparation of vesicles. 

For the LD:CA formulation, purification was carried out using 80:20 hydro-alcoholic solution as a medium for dialysis. The encapsulation efficiency (E%) of CA and DL% of LID in the formulation LD:CA was determined by HPLC (Varian 920-LC Series Liquid Chromatograph) equipped with a chromatographic column C18 reverse phase. The mobile phase used was 0.1% phosphoric acid: acetonitrile (50:50 *v*/*v*) at pH 2.4. The wavelength used for analysis was 280 nm for CA and 210 nm for LID.

The CA E% was calculated using the following Equation (2):(2)$$E\%=\left(\frac{C_{CA\;in\;lidosomes\;}}{C_{CA_{i}}}\right)\times 100$$
where *C_CA in lidosomes_* is the drug concentration measured after the purification process, while *C_CAi_* represents the initial drug concentration. All experiments were conducted in triplicate and expressed as mean ± standard deviation (SD).

### 2.4. Ex-Vivo Permeation Study

Skin permeation profiles of the formulations proposed in this work were evaluated using vertical Franz diffusion cells for 24 h at 37 °C using rabbit ear skin obtained from a local farmer, as reported elsewhere [17]. The skin was frozen in advance at −18 °C and pre-equilibrated in a physiological solution at room temperature for 1 h before the experiments. A portion of skin was placed between the two compartments with the epidermal part in contact with the receptor compartment. The effective diffusion area of cells was 0.416 cm^2^. The donator compartment was charged with 0.3 mL of the niosomal sample in all experiments and covered by Parafilm to prevent water loss, while the acceptor compartment was loaded with 5.5 mL of medium. Specifically, the medium used was distilled water for LD and LD 5.5 B formulations, while a hydro-alcoholic solution (80:20 water:ethanol) for that LD:CA. At regular intervals, the medium was removed to be analysed and restored with equal volume of fresh medium, maintained at a temperature of 37 °C. The amount of LID in the receptor solution was analysed by UV-vis spectrometry, whereas HPLC was employed to investigate the amount of LID and CA released from LD:CA formulation. Skin permeation of free LID and LID gel was also investigated as the control with the same conditions. All experiments were performed in triplicate and expressed as mean ± SD.

### 2.5. Skin LID Retention Studies

The LID amount retained into the skin was evaluated after skin permeation studies. To this purpose, the piece of skin was removed from the Franz diffusion cells, placed in 10 mL of ethanol and magnetically stirred for 2 h. Then, the solution was filtered using 0.22 μm Millipore membrane filters and analysed by UV-vis spectroscopy or HPLC chromatography to evaluate the amount of LID and CA accumulated into the skin. All experiments were conducted in triplicate and expressed as mean ± standard deviation. 

### 2.6. Statistical Analysis

All experiments were performed in triplicate and the results were expressed as mean ± SD. Statistical analysis was performed using a Student’s *t*-test and *p* values of ≤0.05 were considered statistically significant.

## 3. Results and Discussion

LID is a local anaesthetic largely employed for topical therapy and other medical and chirurgical procedures, such as skin sores treatment and wound sutures. Its main use is for parenteral way, typically for intravenous way, instead, less frequent is intramuscular administration [21]. 

The chemical structure of LID suggests the possibility of aggregation in aqueous solution, typical of amphiphilic drugs. In literature, conflicting data are shown; in fact, some authors reported that LID did not lead to the formation of micelles, while in other studies a critic micellar concentration value of 0.12–0.13 mol kg^−1^ was reported [22,23].

For this reason, we decided to investigate the ability of both LIDHCl and LID to form vesicles. Using LIDHCl, we obtained systems that were poorly populated by LD. In fact, vesicular solution appeared only slightly opalescent and the amount of LID retained in the vesicle’s formation was very low. Conversely, the use of LID, already with a quantity of 27 mg but even better with 30 mg, led to obtaining homogeneous vesicular systems with good DL%. The hydration of the lipid film was carried out both in distilled water and in aqueous acid (pH 5.5) or basic (pH 7.9) solution. The size of the LD, depending on the quantity of LID and the hydrating medium used, were between 430 and 574 nm with a PI in the range 0.22–0.27, indicating a homogeneous and narrow size distribution. The negative ζ−potential obtained with LIDHCl may have seemed unusual considering the positive charge of LIDHCl. In order to better investigate the developed formulations, vesicular systems based on both LIDHCl and LID were prepared using for the hydration process solutions with pH equal to the pKa of the drug (pKa = 7.9). In fact, when the pH was equal to the pKa, the drug was present in equal concentrations, both as LID and LIDHCl. The LID obtained, both with the lipid film prepared with LID and with LIDHCl, had a high negative ζ−potential, about −30 mV, and a high DL% that contributed to their formation, over 60%. This could be since only the neutral LID form represented the main constituent of vesicle bilayer, while LIDHCl participated only in a small percentage. Consequently, negative zeta potential values obtained may have been due to the arrangement of amphiphilic molecules with a neutral polar group in their structure, such as that present in LID.

In addition, we investigated the use of LD as drug carriers and used the formulation obtained from 30 mg of LID to load a model drug, CA.

Table 2 reported all characterization data for the formulations prepared.

The morphological analysis showed that LD formulations were spherical, homogeneous in shape and size (Figure 1), with regular and well-defined edges. 

### LD Stability

The stability of LD was evaluated monitoring physical-chemical properties of samples stored at 4 °C for three months and the results are depicted in Table 3. The LD 5.5 B formulation showed a higher size stability than the LD formulation, but no significant changes in DL% and ζ -potential values were observed with either for two months. 

## 4. Ex-Vivo Permeation Studies

### 4.1. Lidosomes

Ex vivo skin penetration efficiency of LD, LD 5.5 B, and LD:CA, was evaluated using the Franz diffusion apparatus. The permeation profile as a function of time of LD and LD 5.5 B samples are shown in Figure 2 and compared with LID solution. As observed, 308 μg/cm^2^ corresponding at 38.38% of drug permeated after 24 h by LD, with respect at almost 214.84 μg/cm^2^ corresponding at 27.89% by LD pH 5.5. However, skin permeations obtained were always higher than those obtained from LID solution, 122.96 μg/cm^2^ corresponding to 5.68%. This suggests that LID, when in the form of vesicular aggregates, act as percutaneous permeation enhancer. This better permeation ability could be ascribed to two different mechanisms. Generally, permeation increases with carrier lipophilicity, probably due to better interaction with the skin layers [17]. Moreover, the skin permeation of amphiphilic molecules depends on their structure [24]. Indeed, as reported in the literature, non-ionic surfactants can interact with both keratin and lipids, altering the lipid layer of the skin and thus making this membrane more permeable [25]. Instead, regarding ionizable compounds, the chemical form that provides better skin permeation is the non-ionised form, as it has a lower polarity and high logP, which provides better affinity for the stratum corneum, which must be crossed to promote absorption after topical application [26]. LID exhibits both behaviours, being an ionizable drug with an amphiphilic structure. Both aspects can, therefore, influence its skin permeation. 

To achieve a localised pharmacological effect, it would be desirable for the drug to accumulate in the upper layers of the skin [27].

In fact, the formation of intracutaneous drug depot would be suitable to prolong drug duration of action over the time leading to better treatment outcomes. Therefore, we decided to investigate the amount of drug accumulated into the skin after skin permeation studies and the results are reported in Table 4. Data showed that LD led to a higher drug depot into the skin, with respect that obtained with LID solution. In particular, the amount of LID retained increased with the pH increase, since the acid-base balance of LID increases the concentration of the neutral species, which are more affinity to skin bilayer.

### 4.2. Gel Formulations and Combined Drug Therapy

Gels have proved to be an advantageous vehicle for drug administration over the skin for their long retention time and ability to slow and prolong drug absorption. Specifically, we decided to investigate the effect of two different gel structure on LD skin permeation, one based on CMC, which does not lead to the creation of structured gels, and one based on pluronic F-127, a well-known pluronic surfactant capable of giving gel-like cubic phase at specific concentrations [28]. 

Ex-vivo skin permeation profiles of CMC LD gel and CMC LID gel are shown in Figure 3A. In this case, skin permeation capacity of LID by LD gel was significantly higher than that achieved by LID gel. In fact, only 11.66% of LID permeated after 24 h from LID-CMC gel. Conversely, a permeation two times higher equal to 20.21% was observed when the drug is in the form of vesicular aggregates, confirming their role as a percutaneous permeation enhancer. Instead, similar performances were observed comparing permeation profile of LD from CMC and F127 gel as shown in Figure 3B. Anyway, a slightly higher amount of LID was retained into the skin with LD-CMC gel compared to that obtained with LD- F-127 (about 450 μg/cm^2^, 353 μg/cm^2^, respectively). These skin depot results were advantageous when an analgesic local effect was required, since the drug was made available in high concentrations for longer times, prolonging drug effects.

Finally, to test the possibility of using LD as a carrier of other pharmacological active molecules, CA was used as a model drug and the effect of CA encapsulation on physical-chemical and permeation properties of LD was investigated. As reported in Table 2, physical-chemical parameters were not affected by CA loading in the vesicular structure and high drug content of both drugs (67% LID DL% and 88% E% CA) were observed. The amount of LID and CA permeated was found to be 89.90 μg/cm^2^ and 126.65 μg/cm^2^, respectively, as shown in Figure 4. Instead, the amount of drug retained in the skin resulted to be about 399 μg/cm^2^ and 20.50 μg/cm^2^ for LID and CA, respectively. Considering that, the proposed system based on the co-delivery of LID and CA could be very interesting topical tools for pain relief able to prolong and improve pharmacological efficacy through synergic effect.

## 5. Conclusions

In this work, aggregation properties of an analgesic drug lidocaine in vesicles (lidosomes) have been investigated using various experimental conditions. Specifically, vesicles were prepared using both lidocaine and lidocaine hydrochloride and different pH in the preparation process. From the experimental data, we concluded that Lidocaine formed vesicle structures with nanometric size, spherical morphology, and good stability at least for 3 storage months. Particularly interesting were the results of the ex-vivo permeation studies that highlighted an enhanced LID skin permeation and a higher drug depot in the skin when in the form of vesicular aggregates with respect to the free drug. 

Moreover, the work evaluated as well as the ability of lidosomes to act as carriers of a model drug, capsaicin, and the obtained results revealed the formation of a reservoir of both drugs into the skin, suggesting the great potentiality of these system to be used in synergic therapeutic treatment of pain relief. However, this paper represents only a preformulation evaluation, and further studies in the future are necessary to investigate the in vivo efficacy of these systems for their consideration as clinically viable formulations.

## Figures and Tables

**Figure 1 pharmaceutics-14-02190-f001:**
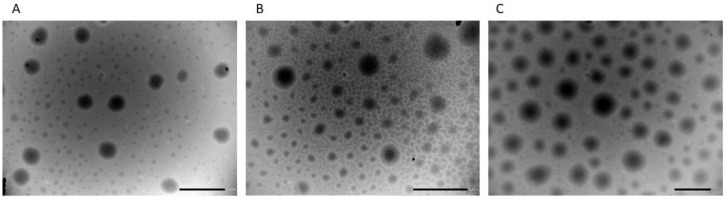
Typical TEM photomicrograph of lidosomes: (**A**) LD 5.5B; (**B**) LD 7.9, and (**C**) LD formulations. Bar is 1 μm.

**Figure 2 pharmaceutics-14-02190-f002:**
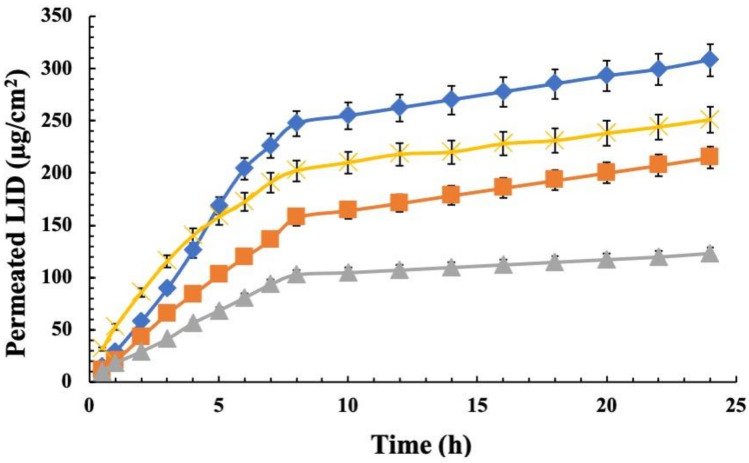
LID permeation profile through rabbit ear skin at 37 °C over 24 h using Franz diffusion cells from various formulations: (**◆**) LD; (**■**) LD 5.5 B; (**X**) LD 7.9; (**▲**) LID SOL. Data are represented as mean ± SD (*n* = 3).

**Figure 3 pharmaceutics-14-02190-f003:**
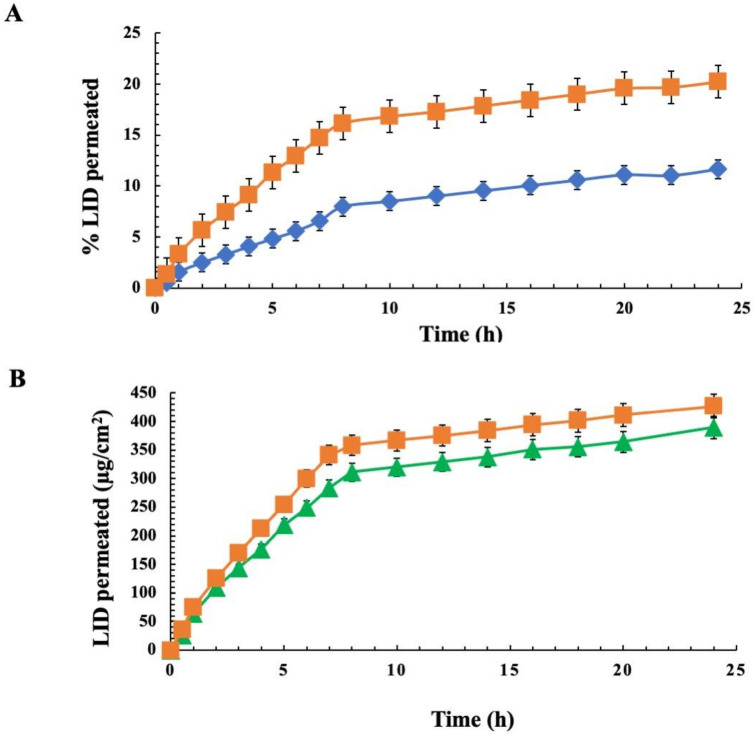
Comparison of LID permeation profiles through rabbit ear skin at 37 °C from several gel formulations: (**A**) (♦) Gel LID vs. (■) LD gel CMC; (**B**) (▲) LD gel F-127 vs. (■) LD gel CMC. Results are expressed as mean mean ± SD (*n* = 3).

**Figure 4 pharmaceutics-14-02190-f004:**
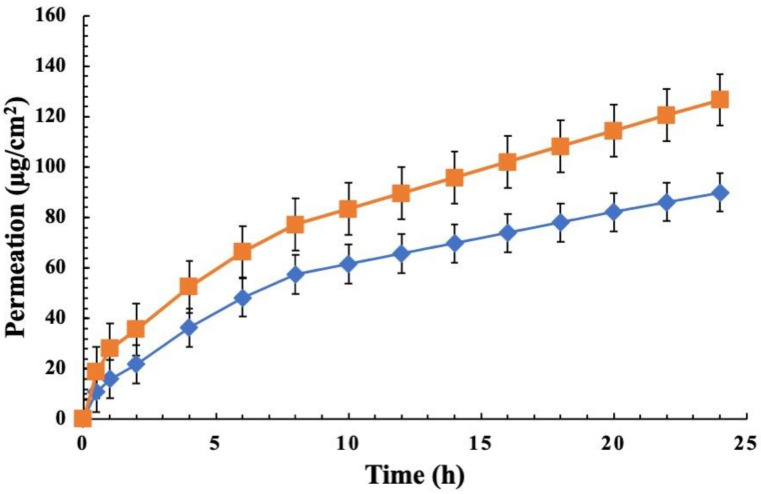
Permeation profile of (♦) LID and (■) CA simultaneously released from LD:CA formulation through rabbit ear skin at 37 °C over 24 h. Results are expressed as mean mean ± SD (*n* = 3).

**Table 1 pharmaceutics-14-02190-t001:** Composition details of lidosomal formulations developed using LID and LIDHCl and various hydration solutions.

Formulation	LID (mg)	CA (mg)	Hydration Medium
LD HCl A	14		H_2_O
LD HCl B	27		H_2_O
LD 5.5 A	27		H_2_O pH 5.5
LD 5.5 B	30		H_2_O pH 5.5
LD	30		H_2_O
LD:CA	27	4	H_2_O
LD 7.9	30		H_2_O pH 7.9
LD HCl 7.9	30		H_2_O pH 7.9

**Table 2 pharmaceutics-14-02190-t002:** Physico-chemical characterization of lidosomes prepared at different concentrations and pH: Hydrodynamic diameter, P.I, ζ−potential, DL%, and E% at 25 °C. Results are the average of three different independent experiments ± standard deviation.

Formulation	Diameter (nm)	I.P.	ζ-Potential (mV)	DL(%) LID	E(%) CA
LD HCl A	707 ± 15	0.249	−13.0 ± 0.709	2.13% ± 0.2	-
LD HCl B	506 ± 12	0.288	−14.3 ± 1.190	1.19% ± 0.7	-
LD 5.5 A	430 ± 10	0.264	−26.1 ± 0.900	19.5% ± 0.5	-
LD 5.5 B	574 ± 11	0.271	−27.6 ±0.833	35.6% ± 0.3	-
LD	437 ± 10	0.229	−23.5 ± 0.208	37.1% ± 0.2	-
LD:CA	519 ± 14	0.277	−23.1 ±0.493	67.5% ± 0.2	87.75% ± 0.8
LD 7.9	512 ± 11	0.287	−31.2 ±0.666	61.2% ± 0.3	-
LD HCl 7.9	612 ± 13	0.277	−30.5 ±0.351	64.8% ± 0.6	-

**Table 3 pharmaceutics-14-02190-t003:** Stability analysis of LD stored at 4 °C evaluated by measuring diameter, P.I., ζ−potential, and DL%. Data was collected at specific time points, up to 3 months, and expressed as mean of three independent experiments ± SD.

Formulations	Time (day)	Diameter (nm)	P.I.	ζ-potential (mV)	DL%
LD	0	437 ± 10	0.229	−23.5 ± 0.208	37.1 ± 0.2
	15	321 ± 9	0.235	−23.2 ± 0.907	36.9 ± 0.3
	30	305± 11	0.295	−22.3 ± 0.819	37.3 ± 0.2
	60	253± 9	0.180	−21.4 ± 0.896	36.7 ± 0.5
	90	175 ± 9	0.294	−19.3 ± 0.451	36.8 ± 0.4
LD 5.5 B	0	574 ± 15	0.271	−27.6 ±0.833	35.6 ± 0.3
	15	570 ± 17	0.204	−27.5 ± 0.173	34.3 ± 0.2
	30	520 ± 19	0.235	−27.4 ± 0.864	34.4 ± 0.3
	60	506 ± 19	0.215	−25.7 ± 0.366	33.4 ± 0.2
	90	501± 18	0.276	−26.5 ± 0.456	33.3 ± 0.2

**Table 4 pharmaceutics-14-02190-t004:** Amount of LID (μg/cm^2^) retained into the skin after 24 h of ex-vivo permeation study carried out for 24 h using Franz cell apparatus.

Formulation	LID Retained into Skin (μg/cm^2^)
LD	387.02
LD 5.5 B	352.16
LD 7.9	545.67
LID SOL	165.87

## Data Availability

The data presented in this study is available in the article.

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
