# Peer review of "Lidosomes: Innovative Vesicular Systems Prepared from Lidocaine Surfadrug"

_pharmaceutics, 2022, doi:10.3390/pharmaceutics14102190_

Round 1

Reviewer 1 Report

Pharmaceutics - Manuscript ID pharmaceutics-1946986

Lidosomes: innovative vesicular systems prepared from Lidocaine surfadrug.  

Dear authors,

The paper is solid and very interesting, and the references used by the authors support the work. I have some suggestions that may improve the paper quality, such as:

- Please, double-check if the best results obtained by the authors have been described in the Abstract section;

- I think the objective of the work can be better addressed, showing what the group intends to present in the paper regarding other works with the same subject;

- Were the data performed in duplicate or triplicate? This information needs to be inserted into the M&M section;

- All abbreviations should firstly be presented with the full description;

- Table and Figure captions should be self-explanatory, i.e., need to contain more description (details) about the conditions and parameters evaluated;

- The Conclusion section needs to be more solid, if possible, point out some perspectives for further experiments in this field;

- Literature references are outdated. Please, update the references in order to cite some papers from 2022.

Author Response

We thank the referee for taking the time to review the manuscript. We have revised the manuscript according to referee suggestion:

Dear authors,

The paper is solid and very interesting, and the references used by the authors support the work. I have some suggestions that may improve the paper quality, such as:

 - Please, double-check if the best results obtained by the authors have been described in the Abstract section;

As suggested by referee we have improved the abstract.

 - I think the objective of the work can be better addressed, showing what the group intends to present in the paper regarding other works with the same subject;

We thank referee for its comment and we have better explained the further step forward performed in this work respect similar ones available in literature.

 - Were the data performed in duplicate or triplicate? This information needs to be inserted into the M&M section;

We thank referee for its comment and we have inserted this information in materials and methods.

 - All abbreviations should firstly be presented with the full description;

As suggested by the referee we have corrected all abbreviations in the text.

- Table and Figure captions should be self-explanatory, i.e., need to contain more description (details) about the conditions and parameters evaluated;

As suggested by referee, we have improved all figure captions.

- The Conclusion section needs to be more solid, if possible, point out some perspectives for further experiments in this field;

The conclusions have been improved as suggested by referee.

- Literature references are outdated. Please, update the references in order to cite some papers from 2022.

We have added several recent references.

Reviewer 2 Report

This research was to study the aggregation of LID in vesicles structure (lidosomes) and to find the best experimental conditions to achieve nanosize carriers with physicochemical properties suitable for topical applications. This study is innovative to a certain extent.

1. The manuscript is well written, but there are a few errors. Please the author needs to review the format of the manuscript carefully.

2.  All tables should use a three-line grid format.

3. The full name should be given for the first occurrence of the abbreviation, and abbreviations should be consistent.

4.  Please number all the formulas.

5. Please add the content of  “statistical analysis”'

6.  Please simplify the language in the "Conclusion" section.

Author Response

We thank the referee for taking the time to review the manuscript. We have revised the manuscript according to referee suggestion:

This research was to study the aggregation of LID in vesicles structure (lidosomes) and to find the best experimental conditions to achieve nanosize carriers with physicochemical properties suitable for topical applications. This study is innovative to a certain extent.

  1. The manuscript is well written, but there are a few errors. Please the author needs to review the format of the manuscript carefully.

We tank referee and we have corrected the text. 

  1. All tables should use a three-line grid format.

Tables have been modified as suggested by referee.

  1. The full name should be given for the first occurrence of the abbreviation, and abbreviations should be consistent.

As referee suggested, we have done it.

  1. Please number all the formulas.

As referee suggested, we have done it.

  1. Please add the content of  “statistical analysis”'

We thank referee for its comment and we have inserted this information in materials and methods. 

  1. Please simplify the language in the "Conclusion" section.

We have improved Conclusion as suggested by referee.

Reviewer 3 Report

The manuscript by Romeo et al describes a system where lidocaine is used as both a drug and a delivery system for the skin.  The basic idea is that the uncharged form of lidocaine will form aggregates that are used as a particulate delivery system.  In addition to the confusing language in many parts of the manuscript, some of the data just don’t make sense, and the authors don’t seem to fully understand the role of the ionized and neutral forms of lidocaine.  More specifically, the experiments involve dissolving the lidocaine at low and high pH, which would dictate the form of lidocaine and presumably the extent of self assembly.  The fact that the formed particles possess a negative zeta potential is not consistent with pure lidocaine particles because lidocaine can only be cationic or neutral.  As such, it is not clear where the negative charge would be coming from.  Taken together, it is not evident that the authors truly understand their system.  More specific comments are below.

1)  The English is hard to understand at times, and I suggest editing by a native English speaker.

2)  The manuscript uses the acronym UVL for small unilamellar vesicles.  Typically SUV is used for this.

3)  The formed particles are much larger than what is typically used for topical delivery, and thus the effect of particle formation is puzzling.

4)  It is claimed that “non-significant” change in terms of size is observed during storage in Table 4, however no statistics were conducted.  The reduction in size appears to be highly significant.

5)  The text in section 4.2 describes data shown in figure 3A, and the description appears the opposite of what is shown in the figure.

6)  Page 8 of the manuscript claims that a much higher skin accumulation is noted for LID even though it’s penetration is lower.  This doesn’t make sense.

Author Response

The manuscript by Romeo et al describes a system where lidocaine is used as both a drug and a delivery system for the skin.  The basic idea is that the uncharged form of lidocaine will form aggregates that are used as a particulate delivery system.  In addition to the confusing language in many parts of the manuscript, some of the data just don’t make sense, and the authors don’t seem to fully understand the role of the ionized and neutral forms of lidocaine.  More specifically, the experiments involve dissolving the lidocaine at low and high pH, which would dictate the form of lidocaine and presumably the extent of self assembly.  The fact that the formed particles possess a negative zeta potential is not consistent with pure lidocaine particles because lidocaine can only be cationic or neutral.  As such, it is not clear where the negative charge would be coming from.  Taken together, it is not evident that the authors truly understand their system.  More specific comments are below.

1)  The English is hard to understand at times, and I suggest editing by a native English speaker.

2)  The manuscript uses the acronym UVL for small unilamellar vesicles.  Typically SUV is used for this.

3)  The formed particles are much larger than what is typically used for topical delivery, and thus the effect of particle formation is puzzling.

4)  It is claimed that “non-significant” change in terms of size is observed during storage in Table 4, however no statistics were conducted.  The reduction in size appears to be highly significant.

5)  The text in section 4.2 describes data shown in figure 3A, and the description appears the opposite of what is shown in the figure.

6)  Page 8 of the manuscript claims that a much higher skin accumulation is noted for LID even though it’s penetration is lower.  This doesn’t make sense

We thank the referee for taking the time to review the manuscript.

1             We have provided  the revision of the text by an English native.

2             We have replaced the acronym UVL with SUV as rightly suggested.

4             We are agree with the referee; the comment is refereed to formulation LD 5.5 B. We had written the 3.1 paragraph

5             The section 4.2 describes correctly the fig 3a, the gel obtained with lidosomes (LD-CMC) shows a permeation higher than the gel obtained with lidocaine (LID-CMC). Probably the label has been confused by the referee. We have increased the label for a better understanding.

6             In the manuscript is reported: “Data showed that lidosomes lead to a higher drug depot into the skin respect that obtained with LID solution.”  In fact, the lidosomal formulation (LD) had showed an increased lidocaine amount retained into skin than the lidocaine solution (LID). Probably the referee has confused the LD and LID formulations. Instead, he finds us completely in disagreement on some of his statements that we discuss below.

3             The referee's doubt is incomprehensible to us. The size of our vesicles (lidosomes) is between 437 and 707 nm. In literature, there are many works that report similar if not greater dimensions. We therefore do not understand the referee's shock. We do not even understand the referee's surprise on the sign of the zeta potential. The zeta potential of a dispersed system, such as ours, is determined by the surface charge density of the aggregates. In fact, the zeta potential is a function of the surface charge which develops when any material is placed in a liquid or better the zeta potential (ZP) refers to the electrical charge at the surface of the hydrodynamic shear surrounding the colloidal particles (Rajagopalan & Hiemenz, 1997). This charge density derives not only from the presence of net charges (for example cations or anions) but also from the distribution on the surface of the aggregates of neutral polar groups (such as esters, ethers, amines). These groups are largely present in all non-ionic surfactants and it is precisely these that give the system a negative charge density. In fact, the niosomal systems made up of these neutral surfactants all have a negative zeta potential, as is widely reported in the literature. Also in our case the systems obtained always have a negative zeta potential that can only be attributed to the neutral form of lidocaine. In fact, the ionic form of the drug has a net positive charge and the dispersed system made from it should have a positive zeta potential. Precisely this observation led us to conclude that lidosomes are formed only from neutral lidocaine.

We observe, in fact, that as the neutral lidocaine concentration varies with the pH, the quantity of lidocaine coming from these dispersed systems increases (DL% in Tab 3).

I hope this clarifies to the referee where the negative charge of the zeta potential comes from, the molecular composition of system and our conclusions.

Most references are reported.

  • Pablo García-Manrique, Noelia D. Machado, Mariana A. Fernández, María Carmen Blanco-López, María Matos, Gemma Gutiérrez, Effect of drug molecular weight on niosomes size and encapsulation efficiency, Colloids and Surfaces B: Biointerfaces, Volume 186, 2020, 110711 (Size emply vesicles 250-1500 nm, non ionic surfactant based niosomes zeta potential -24 mV )
  • Shirsand S, Para M, Nagendrakumar D, Kanani K, Keerthy D. Formulation and evaluation of Ketoconazole niosomal gel drug delivery system. Int J Pharm Investig. 2012 Oct;2(4):201-7. (Range size 4.86-7.38 mm)
  • Luisa Di Marzio, Carlotta Marianecci, Mariadea Petrone, Federica Rinaldi, Maria Carafa, Novel pH-sensitive non-ionic surfactant vesicles: comparison between Tween 21 and Tween 20, Colloids and Surfaces B: Biointerfaces, Volume 82, Issue 1, 2011, Pages 18-24, (non ionic surfactant based niosomes zeta potential -<-40 mV)
  • Parinbhai Shah,Benjamin Goodyear, Anika Haq ,Vinam Puri and Bozena Michniak-Kohn, Evaluations of Quality by Design (QbD) Elements Impact for Developing Niosomes as a Promising Topical Drug Delivery Platform Pharmaceutics2020, 12(3), 246; (Range size 151-919 nm, zeta potential always < 0 mV)
  • ALI, Mohamed, et al. An in vivo study of Hypericum perforatum in a niosomal topical drug delivery system. Drug delivery, 2018, 25.1: 417-425. (size 490 nm non ionic surfactant based niosomes zeta potential -41 mV)

Round 2

Reviewer 3 Report

The arguments you make in your response do not make sense to me.